# Status of Sperm Functionality Assessment in Wildlife Species: From Fish to Primates

**DOI:** 10.3390/ani11061491

**Published:** 2021-05-21

**Authors:** Gerhard van der Horst

**Affiliations:** Comparative Spermatology Laboratory, Department of Medical Bioscience, University of the Western Cape, Bellville, Cape Town 7535, South Africa; gvdhorst7@gmail.com; Tel.: +27-822023560

**Keywords:** sperm, motility, morphology, vitality, CASA, vertebrates, sperm functionality

## Abstract

**Simple Summary:**

In general, wildlife species have been underrepresented, in terms of understanding their reproductive physiology. The artificial propagation of wildlife species, found in aquaculture (e.g., fish) and in protection of endangered species (e.g., black-footed ferret), is pertinent to this discussion. One important approach to addressing this would involve a basic understanding of the structure and function of the gametes of many wildlife species. The focus of this investigation was to provide a better understanding of the physiology of sperm of diverse wildlife species, with special attention given to the assessment of high-quality sperm. Modern approaches using sophisticated microscopy (image analysis) techniques, e.g., computer-aided sperm analysis and sperm flagellar analysis, provided advanced technology to evaluate sperm quality quantitatively. Some of these techniques involved largely automated assessments of many aspects of sperm motility, morphology, vitality, fragmentation and other indirect methods. These modern assessments are fundamental to classify sperm quality. Accordingly, cutting-edge technologies used to define high quality sperm of representative species from most vertebrate animal groups (from fish to primates) were discussed in the present work. These approaches are also important in developing and assessing the best methods to cryopreserve sperm for assisted reproductive technologies in wildlife species.

**Abstract:**

(1) Background: in order to propagate wildlife species (covering the whole spectrum from species suitable for aquaculture to endangered species), it is important to have a good understanding of the quality of their sperm, oocytes and embryos. While sperm quality analyses have mainly used manual assessment in the past, such manual estimations are subjective and largely unreliable. Accordingly, quantitative and cutting-edge approaches are required to assess the various aspects of sperm quality. The purpose of this investigation was to illustrate the latest technology used in quantitative evaluation of sperm quality and the required cut-off points to distinguish the differential grades of fertility potential in a wide range of vertebrate species. (2) Methods: computer-aided sperm analysis (CASA) with an emphasis on sperm motility, 3D tracking and flagellar and sperm tracking analysis (FAST), as well as quantitative assessment of sperm morphology, vitality, acrosome status, fragmentation and many other complimentary technologies. (3) Results: Assessing sperm quality revealed a great deal of species specificity. For example, in freshwater fish like trout, sperm swam in a typical tight helical pattern, but in seawater species sperm motility was more progressive. In amphibian species, sperm velocity was slow, in contrast with some bird species (e.g., ostrich). Meanwhile, in African elephant and some antelope species, fast progressive sperm was evident. In most species, there was a high percentage of morphologically normal sperm, but generally, low percentages were observed for motility, vitality and normal morphology evident in monogamous species. (4) Conclusions: Sperm quality assessment using quantitative methodologies such as CASA motility, FAST analysis, morphology and vitality, as well as more progressive methodologies, assisted in better defining sperm quality—specifically, sperm functionality of high-quality sperm. This approach will assist in the propagation of wildlife species.

## 1. Introduction

It is important to clearly distinguish between sperm quality and sperm functionality. Sperm quality refers to a number of parameters, such as semen volume, pH, sperm concentration, percentage normal sperm morphology, percentage motility and several other parameters (Table 1). A parameter such as sperm concentration provides a basis for what constitutes the characteristics of a typical semen sample, and serves as a reference value for a species. In humans and domesticated species, such values are well-documented [1,2,3]. For example, in many fish species, sperm concentration is on the order of >1 × 10^9^/mL, and sperm motility is above 90% [4]. In contrast, in the naked mole-rat, with a low level of sperm competition, the sperm concentration is 45 × 10^6^/mL and sperm motility is seldom higher than 15% [5]. 

The concern of the present study was how accurately these parameters have been quantified, and whether these sets of standard semen parameters accurately convey the male’s fertility status. Do these parameters predict fertilization success and live birth outcome of a species or the mating system? Let us first review what is well-described in humans and domestic animals before discussing wildlife species. The 5th edition of the World Health Organization’s laboratory manual set out in great detail which semen parameters in humans should be measured to evaluate the characteristics of a semen sample. The manual also included detailed methodologies on how to perform these tests, their variability, and how to rescue them in case the values of duplicates fall outside the accepted statistical range [1]. Despite these efforts by the WHO manual to standardize semen analysis (and while not their intention), the majority of users across the world have used the WHO manual to predict fertility. The problems with this approach have been reviewed [6]. In principle, some of these semen characteristics may relate to the overall quality of the semen but not to sperm functionality or the ability to fertilize. Thus, semen criteria are useful to investigate potential sub-fertility—as good semen quality seemingly relates to the health status of a species. 

In domestic species, similar sets of semen criteria have been constructed to define sperm quality in bulls, boars, goats, rams, horses, chickens and small domestic animals such as cats, dogs and rabbits [2,3,7]. While some of these species have similar characteristics (e.g, sperm concentration and percentage sperm motility), there is a great deal of species specificity, as pointed out earlier. It has also been indicated in many publications that no absolute relationship exists between semen parameters and fertility outcome. One of the earlier investigations showed that in the domestic bull, sperm concentration, percentage abnormal sperm morphology, sperm motility and some measures of metabolic activity were related to conception rate of the semen sample [7]. But even in well-controlled insemination programs, significant relationships among semen parameters and conception rate should not be expected.

It is evident that, in both human and domestic animal species, the determination of reference values for high quality semen is of great importance, particularly when properly quantified. These values serve as guidelines and are potentially useful in insemination programs.

The question then arises which criteria of semen/sperm are more likely to inform us about the functionality of the sperm and potentially the likelihood of its ability to fertilize the oocyte. Table 2 lists an assortment of sperm functional tests, some of which will be discussed in more detail in this work. Superficially, a large overlap between Table 1 and Table 2 may be noted; however, there is a shift in emphasis between them, to use parameters which may distinctly relate to the functionality of the sperm [6,8,9]. For example, laboratories that do not have access to quantitative methods such as CASA (which describes most laboratories) will only be able to make a subjective judgment on the average percentage motility. Even those laboratories that employ CASA will generally only consider average values; however, evaluating sperm motility in terms of CASA sub-populations related to speed and progressiveness, rather than just on the basis of overall population averages, would be more informative. Another crucial aspect of sperm motility is the quantification of hyperactivation due to its relationship to fertilization outcome. Sperm need to hyperactivate (the final stage of capacitation) before they will attach and detach from the oviduct cells or bind to the oocyte [8,9,10]. Moreover, sperm motility is an expression of many aspects of sperm physiology, such as glycolysis, oxidative phosphorylation and membrane intactness, among others. It will be central to the discussions in this review. 

Despite the great advances made in CASA, there are some important shortcomings that make comparisons among taxa difficult. Standardization of a CASA instrument’s configuration settings—e.g., frame rate, particle size, connectivity (in pixels, the next most likely point to find sperm), potential drifting, exclusion of static sperm and cut-off points for sperm sub-populations (e.g., rapid, medium and slow swimming sperm)—is a crucial step that must be taken. Static sperm (e.g., any curvilinear velocity less than 25 µm/s) and particles that may be incorrectly recognized as sperm are two of the most important factors that should be excluded with the appropriate CASA instrument settings, as they can dramatically influence determination of sperm concentration, motility percentages and kinematic parameters. Sperm concentration determination during CASA motility analysis should ideally vary between 10 and 25 million/mL and never exceed 30 million/mL. Higher concentrations may result in too many collisions, affecting both the kinematic parameters and the percentage motility due to motile sperm pushing immotile sperm around (with the latter then being recorded as motile). 

Similarly, the single value of the percentage normal sperm morphology seems to be a simplification that requires more in-depth analysis (such as the different morphology indices). Two of these—the teratozoospermic index (TZI) and the multiple anomalies index (MAI)—relate to the number of abnormalities per sperm or over the entire sperm population. For instance, a species may have a normal percentage morphology but the TZI may be theoretically high (e.g., 2.0), indicating that there were, on average, two abnormalities per sperm. These indices were developed for human sperm [1], but they should be employed for animals, particularly for wildlife and endangered species, with some modifications. While modern CASA systems quantify these indices rapidly, it is time-consuming and inconsistent to assess manually.

From the above-mentioned, it is evident that baseline values are well-established for semen parameters and sperm quality in humans and domestic animals. Nevertheless, such values are not commonly available for most vertebrate wildlife species. It should be noted that some major institutions and entities played major roles in laying the foundation for wildlife sperm research in the 1980s and 1990s, despite not having access to modern CASA systems. There is only space to mention a few in this review, and omission of several others should not be assumed to minimise the important role they played. The Smithsonian in Washington was among the foremost institutions involved; Dr. Wildt, Dr. Bush and Dr. J. Howard played major roles in sperm analysis and cryopreservation, and performed successful insemination of several endangered species (e.g., snow leopard) globally. Dr. Barbara Durrant from San Diego Zoo Global (with her well-known research on the cheetah) and Dr. Bill Holt from the London Zoo were among the first pioneers to test and compare the use of CASA systems on different species, including several wildlife applications.

In this review, the aim was to establish and summarize information regarding semen parameters, sperm quality and sperm functionality in wildlife species represented among the five vertebrate classes (namely, Pisces, Amphibia, Reptilia, Aves and Mammalia). Secondly, the review was intended to ascertain potential relationships and applications of sperm characteristics/functionalities within and between taxa, in captive breeding programmes or aquaculture, in the broader phylogenetic context and in studies on the risk of sperm competition. 

## 2. Pisces (Fish)

The emphasis in this section’s research was on typical bony fish (Teleostei) and not on sharks (Chondrichthyes) or other fish groups. In teleosts, fertilization is predominantly external, and gametes are shed in the water by both the female and male. Teleost sperm are relatively simple in structure, with a small spherical or ovate head, a midpiece with four to five mitochondria arranged in a ring, and a flagellum with the 9 + 2 axoneme arrangement (in most fish). Most teleost species’ sperm do not have an acrosome, and sperm reach the oolemma via micropyles, which are little canals running through the oocyte coverings. However, this is where the similarities between the sperm of freshwater and seawater fish end, in terms of motile life span of sperm. In the former, sperm longevity may be a few seconds, up to 1 or 2 min. In contrast, in seawater species, sperm may last for an hour or longer. 

One classic, remarkable work—relating to measurement of teleost sperm concentration, longevity, osmotic effects, ionic effects, fertilization outcomes and many other aspects in a multitude of species—is Ginzburg’s, originally published in Russian but available in English translation, which is strongly recommended for those who enter the field of teleost spermatology [11]. The very first CASA studies on fish sperm were performed in the mid-1980s and, according to Gallego, more than 170 fish species’ CASA sperm analyses have been reported in more than 700 scientific articles, covering topics ranging from basic physiology to studies on sperm competition and even brood stock management [12,13,14]. 

How have new approaches to assessing sperm functionality contributed to the field? Great advances were made during the last two decades. Two notable research groups—with very comprehensive reviews—include Cosson in 2008 and 2020 [4,15] whose work dealt with marine fish sperm motility, including flagellar analysis, and a number of papers by Gallego and colleagues [12,13,14] focusing on CASA in commercially common marine and freshwater species. This review made it clear that, regarding CASA, fish represent the most-published vertebrate group; in the the last five years, there has been an increase of more than 100% in articles on CASA motility, morphology, vitality and fragmentation. The published works covered a variety of focus areas, from cryopreservation to toxicology and ecology, including environmental factors.

Technique adaptation for capturing sperm of freshwater species for CASA motility analysis is challenging because of the very short duration of maximal sperm motility. For example, in trout, CASA capturing needs to take place within 8 s after dilution with water/medium. Furthermore, the results generated can be influenced by many factors, including the type of CASA system used, recording quality, time of analysis, and frame rate of camera. All of these, if improperly handled, could result in erroneous conclusions [16]. However, despite these challenges, CASA sub-population analysis in salmon could be used to distinguish between mature farmed parr and wild anadromous salmon and ascribe differences to sperm competition (Caldeira et al. 2018) [16]. In many diverse fish species, CASA motility studies have contributed greatly by finding relationships between swimming characteristics like curvilinear velocity (VCL) and sperm quality. It has been shown that percentage sperm motility, progressive sperm motility and kinematic parameters show a high level of correlation with fertilization outcomes and, subsequently, such studies have contributed greatly to brood stock management [12,13,14,17]. Table 3 indicates combined average ranges for the percentage sperm motility and VCL in fresh samples in 13 freshwater species and 5 marine species. There seemed to be similarities in these parameters for the mentioned species.

Few investigations have attempted to use actual captured CASA motility tracks to indicate the presence of different sperm sub-populations and eventually describe typical swimming patterns. It must also be noted that recorded CASA motility tracks are represented as two dimensional images, based on X and Y coordinates. Accordingly, they ignore the Z coordinate. Based on a large body of evidence, it has been suggested that most sperm actually swim in a helix, and many swim in a spherical helix [18]. These authors constructed an Excel program requiring VCL (curvilinear velocity), BCF (beat cross frequency) and ALH (max)(amplitude of lateral head displacement) as input values in order to construct a 3D swimming pattern of a particular sperm. Figure 1A,D and Figure 2D show captured fields of rainbow trout (freshwater fish) and mullet sperm (estuarine but spawning in a marine environment), using the Sperm Class Analyzer (SCA) (Microptic S.L., Barcelona, Spain). However, illustrating the theoretical 3D pattern of the sperm tracks (Figure 1C,E) can be valuable, as it assists the researcher in building a visual picture of the sperm track. Fauvel et al. [19] showed similar analysis fields, as shown in Figure 3 [19], but their research consisted of a single species in different media with the same osmolality. The physiological medium used is a factor that can dramatically change sperm swimming behaviour and interpretation of functionality.

CASA and flagellar analysis assist by providing a better understanding of sperm functionalities. In a recent review (4b), Cosson summarized some theoretical predictions that were made after sperm flagellar analyses of a large number of fish species were investigated. Some important points made are summarized below:The swimming velocity is linearly proportional to the flagellar beat frequency.The shorter the flagellum length, the lower the swimming velocity.The wave amplitude and the swimming velocity are linearly related.Waves of helical shape generate rotation of the sperm cell at a frequency that is in proportion with the flagellar beat frequency, but less efficient than planar waves.

In summary, it seems that a vast amount of quantitative information is available, particularly for fish sperm motility (CASA), and that these many studies have contributed to providing both base line data and many potential applications, relating to cryopreservation, eco-toxicology and in-sperm competition. 

## 3. Amphibia

Amphibians are unique vertebrates, utilizing both the aquatic and terrestrial environments. As such, they certainly have the broadest range of fertilization environments among vertebrates. External fertilization via amplexus (pseudocopulation) is common in frogs (Anura) and may represent release of gametes in water, in foam nests (foam nest frog) and even in damp/wet soil (rain frogs). Many morphological features of amphibian sperm are intermediate between typical aqua sperm (associated with external fertilization) and sperm (related to internal fertilization). It is open to debate whether Anuran sperm fertilization truly constitutes external fertilization, as sperm are virtually deposited directly onto the oocytes, and there is near-immediate contact with the egg coatings after ejaculation. It is undisputed that, at the time of ejaculation, the external environment medium may assist in activating sperm motility. This is a truly intermediate situation between external and internal fertilization—and it is thus unsurprising that many sperm morphological features in frogs are intermediate between external and internal fertilization types. In the other two orders—the legless amphibians (Gymnophiona) and salamanders (Urodela)—fertilization is internal. While a large body of information is available on their sperm structure (most sperm tails have an undulating membrane), limited information exists on sperm functionalities like sperm motility. 

One of the first CASA studies (of several species of most families of South African frogs) was undertaken by Wilson in the mid-1990s [20,21,22]; that work indicated that frog sperm swim very slowly and linearly compared to most other vertebrate groups, with VCL seldom exceeding 80 µm/s and an average speed of about 30 µm/s. In addition, Figure 2D shows star symbol plot analysis, comparing seven sperm characteristics among six South African frog species. Dzminsky [23] made a similar finding and, in contrast with most externally fertilizing aquatic organisms, high fertilization rates were achieved in *C. georgiana*, even with relatively slow sperm swimming speeds. Larozze et al. [24] verified a CASA system for a frog species and found comparable values to the study on South African frogs (and those of others) [25], confirming that frog sperm swim at low velocities as evidenced by the VCL. Apart from using modern technologies such as CASA to investigate amphibian sperm functionality, efforts toward successful cryopreservation and the development of global sperm banks need to be accompanied by detailed quantification of amphibian sperm parameters and functionality at many levels—including molecular biology approaches such as proteomics and metabolomics [6]. 

The next three examples of diverse frog species’ sperm denote the importance of using different sets of sperm parameters, e.g., sperm morphology and motility or site of sperm collection, to better understand their sperm biology and fertilization environment.

In the foam nest frog, sperm are ejaculated simultaneously by several males in a dense foam nest containing ovulated oocytes. The sperm of this species have 11 tight head helical coils, and the sperm head apparently unleash and stretch out to several times their length (Figure 2A–C). This change in head morphology presumably assists sperm in reaching the egg—and explains a mechanism used to get to the oocyte in view of poor motility/immobility. The sperm of the foam nest frog is hardly capable of progressive motility [18,19]. Furthermore, in various South African frog species, including the foam nest frog (in which head length exceeded 20 µm), sperm were either immotile or achieved only slow swimming speeds, and seemed to be associated with terrestrial breeding [18,19]. 

A large divergence in sperm quantity, motility and sperm length among *Pseudophryne guentheri* populations seemed to correspond with annual rainfall and rainfall seasonality [25]. The authors of a study on the topic suggested that these xeric and mesic populations provided a mechanism for cryptic speciation in this widespread Australian genus. Sperm collected from amplectant *Xenopus laevis* (African clawed toad) swam faster than sperm collected from the testis, as measured using CASA [20].

It should be emphasized that it is often the combination of sperm functionalities/parameters that provide better insights in sperm biology and its relationships in terms of fertilization environment and sperm competition. In the above three examples, motility, morphology, environmental conditions and sperm origin/maturation were very important factors. 

In summary, most representatives of the three amphibian orders have cigar-shaped or filiform sperm heads, a small midpiece and, in many species, a tail consisting of the 9 + 2 axoneme with an undulating membrane and axial rod attached. Sperm of most Anuran amphibians (frogs) are extremely slow swimmers (as measured by VCL) and, at least in frogs, sperm is deposited almost directly on the eggs in the process of pseudo-copulation (amplexus). 

## 4. Reptilia (Reptiles)

Reptiles, birds and mammals are collectively called Amniota, because they share membranes (amnion, chorion and allantois) for embryo development. Reptiles constitute four orders: the turtles (Chelonia); lizard-like animals (Rhynchocephalia), of which the tuatara is the only living member; snakes and lizards (Sauria or Lacertilia); and crocodiles (Crocodilia). At least one example from each of these orders was selected for discussion of sperm/semen parameters, with special reference to CASA and other new technologies which shed light on sperm functionalities. Over the last two decades, in particular, a greater awareness of the need to protect endangered and vulnerable species led to enhancements in technologies associated with semen collection, cryopreservation and insemination in reptiles and skinks (lizards) [24,25,26]. 

Some common sperm traits among reptiles include a long thin head, a very long midpiece, relatively slow swimming speeds (VCL < 100 µm/s and typical range of 60 to 80 µm/s) and a linear swimming pattern (LIN of 60 to 80%). A unique aspect of snake (and most other reptile) sperm is the extremely long midpiece, which may reach 111 µm in *Waglerophis merremii*, a non-poisonous snake from Argentina [25]. The tuatara (genus Sphenodon) possesses extremely long sperm, with typically spiraled-shaped heads spanning over 50 µm, and their sperm swim linearly (based on personal observation). It has been suggested that longer sperm (particularly a longer tail) would be advantageous in sperm competition, but that may not necessarily be the case in all reptiles. In two snake species with very diverse reproductive patterns, the percentage “sperm motility and swimming velocity respond mainly to environmental conditions imposed by mating systems rather than to selection by sperm competition” [27].

In turtles, recorded sperm motility values were unexpectedly higher at reduced temperatures in two of the three species examined [28]. The species with increased sperm motility at reduced temperatures could be those which have participated in fertilizations with stored spermatozoa; their CASA kinematics and motility results supported this view [28].

Crocodilian (caiman) sperm motility [29] has been studied with special reference to sub-populations for rapid progressive, medium progressive, rapid non-progressive and slow-non progressive percentages. It was comforting to find that more investigators—such as the aforementioned—have used the sub-population approach, providing information on sperm functionality rather than just overall averages. Furthermore, in four caiman ejaculates, the VCL was low at about 50 µm/s and, in contrast to snakes and lizards, the linearity and progressive motility were also low. 

Friesen et al. [30] performed an ingenious set of experiments in the painted dragon lizard, correlating telomere length to many aspects of reproductive features in both males and females, but specifically in relation to sperm traits. Telomere length correlated positively with both age and fertility. They discovered with inverse relationships that sperm telomere length was highly correlated with kinematic parameters such as VCL. Their study incorporated important new approaches to studies of sperm functional features and related it to other non-reproductive parameters using sophisticated molecular biological approaches such as PCR [30].

In summary, reptile sperm are among the longest of any vertebrate group, and they are typically recognized by the filiform head, very long midpiece and relatively slow swimming sperm. Studies have revealed some surprising relationships between mating systems and sperm competition among reptiles, and it is comforting to know that more quantitative approaches (such as CASA) have been used to quantify sperm functionality with many applications.

## 5. Aves (Birds)

The two main groupings (or superorders) of birds are the Paleognathae—flightless birds (seven orders), e.g., the ostrich, emu, rhea, and kiwi—and the Neognathae, including all flying birds except for penguins (24 orders). The sperm of most birds are filiform like those of reptiles but the midpiece is shorter and often contains one or two (and in penguins, about eight) mitochondria [31]. The principal part of the tail has a simple 9 + 2 axoneme and, in some species such as African and Rockhopper penguins, there may be more than one axoneme inside one cell membrane in a small percentage of sperm. Are there major morphological differences in the sperm of the two superorders? The sperm head of Neognathae—and specifically passerine birds—are often helically shaped, though the extent of this varies among different species. In contrast, Paleognathae typically have more simplified, filiform sperm heads. 

A large number of papers relating to sperm selection, sperm competition and quantitatively measured sperm traits have appeared in the last two decades. These mainly focused on passerine birds, and it would be a formidable task to cover these even superficially. Instead, select important papers, relating to sperm traits and sperm functionality in select passerines, penguins and at least one paleognath bird (the ostrich) were selected for discussion.

A major paper, including 47 passerine bird species, reported a positive correlation between extra paternal frequency and sperm swimming speed, indicating that sperm swam faster when there was a higher risk of sperm competition. However, clutch size was negatively correlated with sperm swimming speed, illustratingthe significance of both sperm competition and clutch size as evolutionary driving forces for sperm swimming speed [32].

Another related paper revealed a positive association between sperm head morphology and sperm swimming speed. Those species with “large sperm and a strong helical form, a more pronounced waveform along the cell core, and a more pronounced helical membrane had faster swimming sperm”. Furthermore, it appears, from this study, that sperm head morphology may influence sperm physiology in songbirds [33]. 

It was demonstrated in two important papers [34,35] that Zebra finch males selected for long sperm and showing higher velocities (CASA determination) had a competitive advantage in reaching the eggs, when compared to males with shorter and slower sperm. In the second paper, an anomalous situation was found; males with longer sperm had shorter midpieces but higher ATP levels.

A large body of information has been gathered on sperm traits in many different penguin species. In the African penguin (*Spheniscus demersus*) there are two breeding periods per year: a short breeding period from January to February, and a longer one beginning in June and ending in October. When sperm traits such as ejaculate volume, percentage motility and sperm kinematics were compared over these two breeding periods, no differences were found [36]. The first season seemed to have been dominated by much higher blood steroid concentrations in both males and females than the second longer breeding period, which may suggest accelerated sperm maturation in the short breeding period [34]. 

The first quantitation of ostrich sperm parameters involving CASA and the effect of several abiotic factors on sperm biology was performed by Bonato and colleagues [37]. They found that “body temperature and slightly alkaline conditions seem to stimulate ostrich sperm motility, but it appears that an acidic to neutral pH range for the ostrich-specific diluents is required to ensure better sperm survival for artificial insemination during in vitro storage”. These results paved the way for cryopreservation and insemination in this species [37].

Figure 3 includes random but representative CASA-captured fields with six kinematic parameters using the same diluents (Ham’s F10) for Zebra finch and ostrich, in order to provide some idea of the variation in CASA results among these two extreme bird species. For Figure 3, the red tracks show rapid progressive sperm, the green tracks represent medium progressive sperm and blue tracks represent slow or non-progressive sperm. Zebra finch sperm have considerably slower swimming speeds than ostrich sperm and Zebra finch sperm swim along a very straight trajectory—to the extent that VCL, VAP and VSL values are similar. Furthermore, ostrich sperm swim rapidly but with considerably lower progression than Zebra finch sperm, particularly when the kinematic parameters (such as VCL) of the rapid swimming population are considered. The importance of the sub-population approach has been emphasized throughout this paper, as it is intended to provide better clarity on those sperm populations most likely to fertilize the oocytes and, accordingly, is related to sperm functionality in a species. After all, it is likely that the rapid population of sperm will reach the oocyte first. 

In the past it has been suggested that bird sperm does not seem to exhibit hyperactivation, a crucial step in capacitation, particularly of mammalian sperm. However, it is relatively easy to induce such capacitation changes (e.g., hyperactivation) in the ostrich (based on personal observation). Figure 3 (4) shows a sperm track that resembles hyperactivation in the typical mammalian situation. It is aspects such as capacitation changes that are sorely under-represented or missing in bird spermatology; these will be elaborated under mammals.

It may be of interest to establish whether sperm from Paleognathae such as the emu and kiwi have comparable sperm traits to that of the ostrich, in order to explore potential phylogenetic considerations. Furthermore, is there a clear distinction between sperm traits of Paleognathae and Neognathae? Questions relating to body size, sperm velocity, length of the female reproductive track, sperm size, kinematics, sperm concentration, sperm number per ejaculate and mating systems are foremost among possible future investigational paths.

## 6. Mammalia (Mammals)

Mammal orders vary enormously, and are arranged under three infraclasses: the egg-laying monotremes, the marsupials and placental mammals. The latter group constitutes about 16 orders and will be the focus of this review. Although it is not possible to show and compare examples from even a fairly representative group of these orders, considerably more progress has been made to study sperm functionality in mammalian wildlife species than in any other vertebrate group. 

Since many studies on wildlife mammalian species’ sperm functionality have incorporated different aspects of capacitation, some clarification is required regarding this physiological process. Sperm cannot fertilize the oocyte unless they become capacitated. Capacitation takes place in the female reproductive system and involves several dozen steps. In brief, sperm are stripped from their seminal protein coverings while traversing the cervical mucous. During this process many receptors are exposed; after, most of the capacitation processes take place in the oviduct. One of the major events is that Ca^2+^ ions enter the sperm cell. This is required for a landmark event, referred to as hyperactivation. Hyperactivation is required for three reasons: to help sperm to attach to the cells of the oviduct, to detach from the oviduct cells when the oocyte arrives, and finally, to penetrate the oocyte. Ca^2+^ is also required for sperm to undergo the acrosome reaction. Sharon Mortimer, in 1997, alluded to the physiological importance of sperm motility analysis in mammals, with emphasis on the significance of hyperactivation and the use of CASA for quantification [8]. Hyperactivation has subsequently been well-quantified in humans, while work is in progress to do the same for several wildlife species. 

For sperm traits/sperm functionalities in mammals, the discussion will use orders as the framework, since there seem to be many sperm species similarities within orders. 

### 6.1. Carnivora

Carnivora is one of the orders in which major sperm functional research has been performed, probably because this diverse order had several species on the IUCN Red List of Threatened Species. In fact, one of the species studied was the black-footed ferret (BFF), which was thought to be extinct. 

An extensive research group, led by Dr. Bob Atherton, studied and cryopreserved sperm of the nine founder males of this species in Wyoming in the late 1980s and early 1990s as part of a captive breeding program (Figure 4A,B). In parallel, sperm from Siberian ferrets, domestic ferrets and Fitch (hybrid) ferrets were studied to investigate techniques applicable to the BFF—these three species were found to be satisfactory surrogates for BFF if required [38]. CASA was applied for the first time, testing the effects of various extenders on sperm motility of all the above ferret species, including the nine founder males. Sperm kinematics did not differ among the four species but huge differences were evident for sperm of all species when different physiological media were employed (Figure 4C,D). Initially, a large percentage of sperm morphology abnormalities were found in BFF when compared to Siberian ferrets (Figure 4E,F). The Smithsonian researchers were the first to produce BFF offspring after uterine insemination via laparoscopy [39]. The captive breeding programs were well-coordinated by several prominent zoos in the USA. Some BFFs were subsequently reintroduced into the wild over the last three decades, and have been breeding successfully under natural habitat conditions. However, the extent to which inbreeding has occurred is of concern.

The Comparative Spermatology Group at the University of the Western Cape, in conjunction with several other research groups, including the National Zoological Garden in Pretoria as well as Dr. Imke Lueders from Hamburg, studied lion semen and sperm functionality in some 50 male lions in South Africa (some aspects published [40], but mostly unpublished). Urethral catheterization (UC) was used for the first time in a wildlife species, after adapting the technique for domestic cats, to collect sperm from African lions [38]. The advantage of the UC protocol was that only very light ketamine/medetomidine anaesthesia was required, and sperm was collected uncontaminated from urine, while the stress normally caused by electro-ejaculation (EE) was avoided. 

The lion research paved the way to successfully collect and analyse sperm from the African leopard, cheetah, Bengal tiger and even the wild dog. In all these cat and dog species, there appeared to be three commonalities. First, they seemed to have a fairly high percentage of abnormal sperm, and second, sperm concentration was typically more than >2 × 10^9^/mL in these species. Third, CASA revealed species-specific kinematic traits, but otherwise, these species had similar sperm motility parameters. Their sperm tracks typically exhibited mostly progressively motile sperm in a medium such as Ham’s F10, and the actual values for the kinematic characteristics fell in a narrow range. Figure 5 shows representative sperm tracks and kinematic values for rapid progressive sperm of the African lion, Bengal tiger and African leopard. Progress has also been made to adapt different domestic animal settings and establish the correct CASA settings for sperm motility (for example, progressive motility) for the endangered Arabian leopard [41]. 

The challenge now is to apply more quantitative sperm functional approaches to improve testing of sperm cryopreservation protocols and the insemination of functionally high-quality sperm for the artificial propagation of those cat species that are on the endangered species list. Mackie et al. [42] experimented successfully using UC in the BFF and showed that it was not only less invasive than EE but that semen of good quality could be retrieved.

### 6.2. Proboscidea

Both Indian and African elephant sperm have been studied in considerable detail. Luther et al. [43,44,45] performed the first in-depth study on the semen quality—using CASA and sperm functionalities such as hyperactivation—in free-ranging (wild) African elephants from the Phinda Private Game Reserve, close to Kruger National Park. CASA, involving sperm concentration, motility, morphology and vitality, assisted greatly in establishing that African elephants have high sperm quality throughout the year, but that sperm functionality is highest during the rainfall period, when most cows are in oestrus (Figure 6B–E). In addition, one of the important genes in hyperactivation, CatSper 1, was confirmed in the African elephant, helping to explain the high percentage of hyperactivation in African elephant bulls, particularly when they are in musth (Figure 6A–C). 

The sperm of these elephants were cryopreserved and successfully inseminated in elephant cows in zoos in Europe, and calves were produced, with the advantage of introducing new genetic material [43]. While the African elephant is threatened in some African countries, it is not the case in South Africa. In fact, some of the national parks experience overpopulation and some destruction of flora. 

The above investigations were followed by an in-depth study on male contraception in African elephants in South Africa [46]. Two to three injections of a highly affordable and common anti-GnRH contraceptive used on boars in Europe and South Africa (Improvac, Zoetis, Sandton, Johannesburg, South Africa) was administered by dart over a period of several months. These administrations resulted in a significant decrease in the sperm count, an increase in abnormal sperm heads (often with detached tails) (Figure 6F) and finally, 100% contraception. Since testosterone levels decreased significantly using this approach, it can also be used as a strategy to decrease aggression in problematic African elephant bulls in smaller wildlife parks. Reversal was possible when the contraceptive was withdrawn from adults but not when it was injected into juveniles. 

Another study (in progress, with some of the data published as a conference poster) is related to the phylogenetic relationships of African Elephants, hyraxes and manatees [47]. In this study, the focus was on similarities and differences among the three species’ sperm characteristics. The idea was to establish if the African elephant was more closely related to the manatee or to the hyrax based on sperm traits. Earlier work on the anatomy of the hyrax and the elephant seemed to show a very close relationship. Initial data suggests, however, that more sperm characteristics are similar between elephants and manatees (e.g., sperm morphometry) than elephants and hyraxes. Additionally, in elephants, the lung coverings (apparently, not true pleurae) are closer to that of marine mammals, suggesting that elephants might have secondarily emerged from a water environment.

### 6.3. Rodentia 

The order Rodentia exhibits the greatest interspecific variability of any mammalian taxa, and accounts for the highest species richness of all mammals [48]. Their sperm are complex, and show considerable differences in terms of head shape (e.g., ovoid to falciform, with one or many apical hooks). Total sperm length can range from as small as 35 µm to 258 µm. The focus of wildlife rodent sperm research has predominantly been morphological studies describing the complex structure of the sperm, relating it to phylogenetic/taxonomic affinities and the risk of sperm competition. In this context, numerous papers have been published by many authors [49,50,51,52]. While these authors have deciphered many different sperm traits [51], they have not covered sperm functionality (the focus of this review) as comprehensively as in many of the other mammalian orders discussed. An exception is the recent research on the capybara [53], analysing a whole range of male reproductive parameters before and after treatment with an anti-GnRH contraceptive. Very detailed CASA of some fourteen kinematic parameters and sperm sub-populations showed the contraceptive effects with great clarity. In contrast, acrosome integrity and mitochondrial activity showed no significant differences between the control and treatment groups, while large testicular volume and testicular histological differences were evident. Tannenbaum [54] used CASA to study sperm parameters of a range of rodent species in potentially contaminated sites, and aimed to derive the health status of these animals. Accordingly, good quality sperm parameters related to better health status. This is an important application of sperm functionality—using it in order to establish health status in a routine manner.

### 6.4. Artiodactyla

Even-toed ruminants—including deer and antelope-like mammals—represent an important order, with many studies focusing on their sperm biology, both in general and more specifically in terms of sperm competition and cryopreservation. Sperm characteristics of red deer and antelopes have received significant attention. Most investigators used a combination of a CASA sperm motility sub-population approach and other sperm functionality tests, such as vitality and the hypo-osmotic swelling (HOS) test. Martinez-Pastor [55] analysed semen parameters in Iberian red deer and roe deer for osmolality, pH, motility (subjectively and with CASA), HOS test reactivity, acrosomal status and viability (assessed with propidium iodide). The focus of this study was on how long sperm functionality could be maintained after collection. It provided important information on sperm viability, for AI purposes, for example.

It has been shown that sperm design and velocity play key roles in influencing sperm performance and, therefore, that they can determine fertilization success. An important approach in this investigation was to investigate sperm sub-populations, which, based on sperm morphometry and velocity, showed a relationship with fertility. The results indicated that males with high fertility rates had fast, linear-moving sperm, and could further be correlated with specific sperm morphology populations [56]. 

Cauda epididymal sperm, collected post-mortem from three antelope species (blesbok, impala and springbok) were analysed, fresh and cryopreservation, using CASA. Fourteen sperm motility characteristics were studied. The strength of the study was evident in the sub-population approach; for example, sperm were categorized as rapid and rapid progressive or non-progressive. This aided greatly in the understanding of those populations of sperm that are most likely to fertilize the oocytes. The interspecies comparison of the kinematic parameters of sperm among the antelopes over several end points contributed to a better understanding of comparative sperm physiology. This forms an important step “in the development of species specific assisted reproductive protocols and techniques (ARTs) for ex situ conservation of these species” [57].

### 6.5. Odontoceti

Despite the problems in obtaining sperm from any marine mammal, great progress has been made with dolphins, belugas, killer whales and manatees over the past two decades. Extensive sperm research on several dolphin species (with special reference to bottlenose dolphins) has been performed, with pioneering work by Robeck and Obrien [58]. They developed sperm cryopreservation and artificial insemination methods for a number of cetaceans and, in most instances, fresh ejaculates were high-quality, with particularly high motilities and low levels of DNA fragmentation. 

Ruiz-Diaz cooled dolphin sperm to 5 °C instead of freezing it, and the sperm still showed a high degree of viability and swimming capacity when maintained in Beltsville Thawing Solution and measured by CASA. Maintaining sperm at 5 °C (refrigeration) instead of freezing provides an alternative storage method, on the condition that the sperm are used within 5 to 7 days [59], an approach similar to that used in the boar industry.

In a further study on bottlenose dolphins, very similar results to previous studies on CASA motility were found, and the study was extended to assess sperm sub-populations. In this regard, rapid sperm populations were quantified in Hams’ F10 and served as an important baseline and yardstick for future investigations [60]. In addition, the sperm functionality (as measured using CASA motility) was higher in second and third ejaculates. Accordingly, it may be useful to pool semen from these ejaculates for freezing and insemination after thawing, or even for inseminating fresh sperm.

In a recent investigation, Cowart et al. studied sperm motility of fresh and consecutive ejaculates of manatee sperm [61]. In ejaculates two to five, sperm quality was high, as measured using CASA. In addition, high levels of normal chromatin condensation and chromatin maturation were reported, and almost 80% of manatee sperm had intact acrosomes [61]. 

### 6.6. Primates

Several studies compared sperm traits in the large apes and humans. Anderson’s study clearly showed that there are distinct differences between human and chimpanzee sperm characteristics—such as JC-1 staining of mitochondria. This has been supported by flow cytometry [62]. Mating systems seemed to explain most of the reported differences; chimpanzees exhibited higher sperm numbers and higher levels of sperm energetics than humans. Furthermore, gorillas, gibbons and orangutans seemed to be more similar to humans, showing lower levels of sperm competition than Panids such as chimpanzees. One review article showed what happens to sperm traits during the absence of competition. The authors commented on sperm traits and lower levels of sperm competition in humans and gorillas [63]. 

Maree showed similar trends when using CASA, sperm energetics and mitochondrial structure (using transmission electron microscopy) to compare human sperm with those of three Old World monkey species, namely the vervet monkey, the rhesus monkey and the chacma baboon [64]. For all sperm traits, the monkeys showed sperm features linked to intense sperm competition, compared to humans with a low level of sperm competition. See Figure 7B–E, referring to mitochondrial activity in these species, as measured by Mitotracker Red. In addition, flagellar analysis of vervet monkey sperm represented a new development in the study of sperm functionality (Figure 8 and explanation of FAST in the section below). FAST analysis shown in Figure 8 is based on principles as outlined by Gallagher [65] and methodology applied for vervet monkey based on van der Horst [66].

## 7. Further and Future Sperm Functional Applications in Wildlife

Acrosome intactness and acrosome reaction are factors used to quantify sperm quality as part of a battery of sperm functionality tests. Figure 7A also shows FITC-PNA fluorescence analysis of Tankwa goat sperm, revealing a high percentage of acrosome intact sperm (indicated by green fluorescence). It was shown that—in this feral goat species that has had no human interference for 70 years—80% to 95% of sperm had intact acrosomes.

Maree showed the importance of employing the mitochondrial membrane potential as a sperm functionality test [64]. In this instance, Mitotracker Red was used, in both fluorescence microscopy analysis and quantification using flow cytometry, to compare human sperm to that of three non-human primate (NHP) species (Figure 7B–E). Human sperm has only about 27 mitochondria, compared to that of the three NHP species which have about 100 mitochondria per sperm. Mitotracker Red is a measure of mitochondrial activity and vitality. In the above comparison, the three NHP species showed a higher level of fluorescence due to greater mitochondrial activity. There was also a significantly higher fluorescence (as quantified using flow cytometry) than was observed in human sperm. Mitochondrial fluorescence and midpiece volume also indicate a large difference between species where there is a high risk of sperm competition, compared to humans who, as a predominantly monogamous group, have a low risk of sperm competition. Accordingly, the Mitotracker quantification of the sperm of wildlife species may be useful as a further confirmation of the sperm quality, and could also serve as an important method to establish the risk of sperm competition. 

In summary, sperm of species from only 6 diverse mammalian orders have been discussed. In most instances, sperm traits and sperm functionalities of species within an order show greater similarities than between orders. 

## 8. Summary/Discussion of Some Sperm Traits of the Five Vertebrate Classes

Many of the advanced CASA technologies used to study and evaluate sperm function in humans and domestic animals are now commonly applied in wildlife species, including several mammalian orders. This is important in view of the captive breeding of endangered species and the preservation of the gametes and embryos at very low temperatures. In addition, technological developments in the field of CASA have become user- and field-friendly, which means that sophisticated analysis can be performed under almost any condition. Table 4 has been constructed to summarize and discuss some of the sperm traits that help to characterize and evaluate sperm quality and sperm functionality in the five vertebrate classes. The emphasis in this review was on computer-aided sperm analyses. In most vertebrate classes, sperm concentration is high and often exceeds a billion sperm/mL. It is important to evaluate sperm quality/functionality in a broader context, rather than just dissecting the values of the different sperm traits. Accordingly, many factors—including phylogenetic position, fertilization environment, environmental factors and sperm competition—need to be considered. Often a lack of sperm competition is associated with a low sperm concentration, as is the case in the naked mole rat. Conversely, in most species, a high level of sperm competition is evident, and is associated with sperm traits including high sperm concentration, a high percentage of morphologically normal spermatozoa, and a high percentage of motile sperm that are swimming rapidly and progressively forward. These traits are typically associated with a high percentage of live sperm (vitality) that have a high linearity and a low level of DNA fragmentation (Table 4). 

Which sperm traits distinguish a particular vertebrate class from the others? Clearly the teleosts (bony fish) are mostly broadcast spawners. They have considerably higher sperm concentrations than the other classes to compensate for the water dilution effect. Structurally, their sperm have a very simple form compared to the very complex design of internal fertilizers such as reptiles, birds and mammals. The design in this case is to withstand the viscous forces during motility, whereas the sperm of teleosts swim in an aqueous environment. 

Amphibians occupy an intermediate position with pseudo-copulation and, during amplexus, eggs are released in close proximity to the male cloaca and sperm are deposited directly on the eggs. It is thus not surprising to find that CASA analysis revealed very slow-moving sperm and, in some instances, very poor motility among amphibians (Figure 4).

Some species, particularly the carnivores, exhibit a relatively low percentage of normal sperm morphology (about 65% in ferrets; [38]) when compared to many other mammalian species. Sperm traits need to be viewed in this overall context before defining sperm quality. 

CASA and its related technologies (e.g., measurement of mucus penetration and hyperactivation) are useful in objectively defining high quality and functional sperm, since they largely relate to the ability of a sperm to fertilize an oocyte. These measurements rely on a combination of kinematic values with specific cut-off points that are species-specific. Finally, these and many other sperm functional measurements cannot be performed manually and are dependent on CASA technologies. 

## 9. Conclusions

In this review, the aim was to document and summarize information regarding sperm quality and sperm functionality in wildlife species among the five vertebrate classes. This was a daunting task, considering the large species diversity in vertebrates. Despite all the pitfalls, the examples used provide a snapshot of their sperm traits. There seem to be sperm-specific traits for external fertilizers (teleost and anuran amphibians), as compared to internal fertilizers. These traits largely derive from adaptations to the fertilization environment. It was emphasized, however, that care needs to be exercised in defining sperm quality without taking many other factors (e.g., sperm competition, phylogenetic relationships and breeding season) into consideration. Finally, most of the documentation on sperm traits in wildlife species—often made without proper quantification by objective, repeatable methodologies such as CASA—is subject to error and not reliable.

## Figures and Tables

**Figure 1 animals-11-01491-f001:**
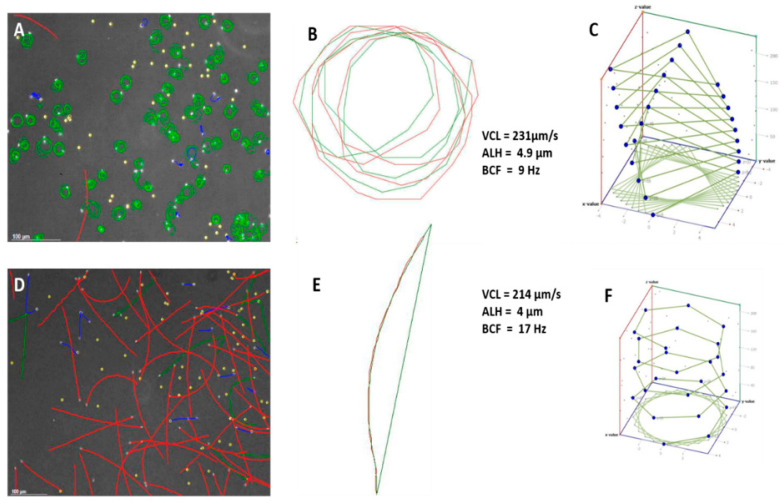
Sperm tracks after CASA capture and track 3D reconstruction. (**A**) One field of rainbow trout (freshwater) sperm captured using SCA software (Microptic S.L.). The sperm swim in a tight helix. Red tracks = rapid progressive; green tracks = rapid non-progressive; blue tracks = slow non-progressive. (**B**) Detail of one sperm track reconstructed from X and Y coordinates every 100th of a second. (**C**) Three-dimensional reconstruction of track in B using the method of van der Horst and Sanchez [16]. (**D**) One field of mullet (seawater) sperm captured using SCA software (Microptic S.L.). Most sperm swim progressively forward. (**E**) Detail of one track reconstructed from X and Y coordinates every 100th of a second. Red tracks = rapid progressive; green track = rapid non-progressive; blue tracks = slow non-progressive. (**F**) Three-dimensional reconstruction of track in B using the method of van der Horst and Sanchez [16]. VCL = curvilinear velocity; ALH = amplitude of lateral head displacement; BCF = beat cross frequency.

**Figure 2 animals-11-01491-f002:**
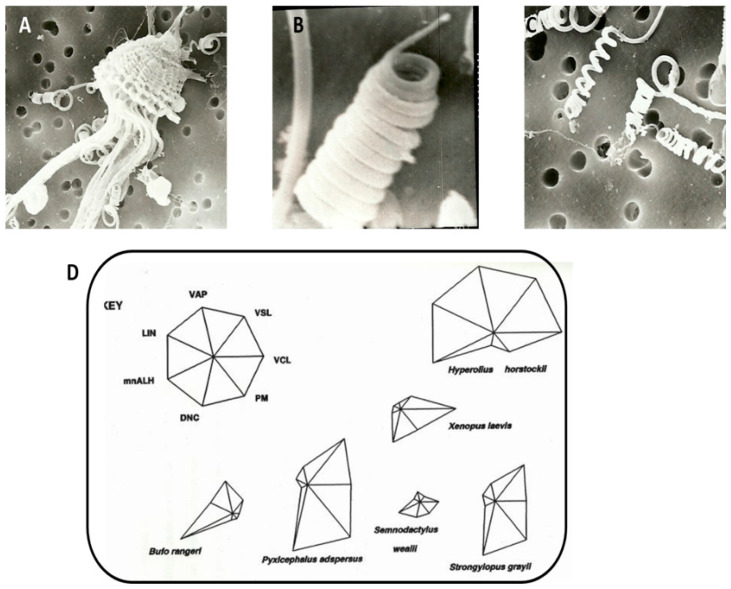
Scanning electron microscopy of the foam nest frog (*Rhacophorus* sp.). (**A**) Bundle of sperm associated with one Sertoli cell in testis. Note the head with about 11 tight coils. (**B**) Higher magnification of one sperm showing detail of acrosome and the 11 head coils. (**C**) Extended head coils once sperm are exposed to external medium and (presumably) foam produced by female. (**D**) Star symbol plot analysis on the basis of seven kinematic parameters comparing sperm motility of six frog (Anura) species from South Africa. Abbreviations for kinematic parameters: DNC, DANCE; LIN, Linearity; mnALH, Mean Amplitude of Head Displacement; PM, Progressive motility; VAP, Average path velocity; VCL, Curvilinear velocity; VSL, Straight line velocity.

**Figure 3 animals-11-01491-f003:**
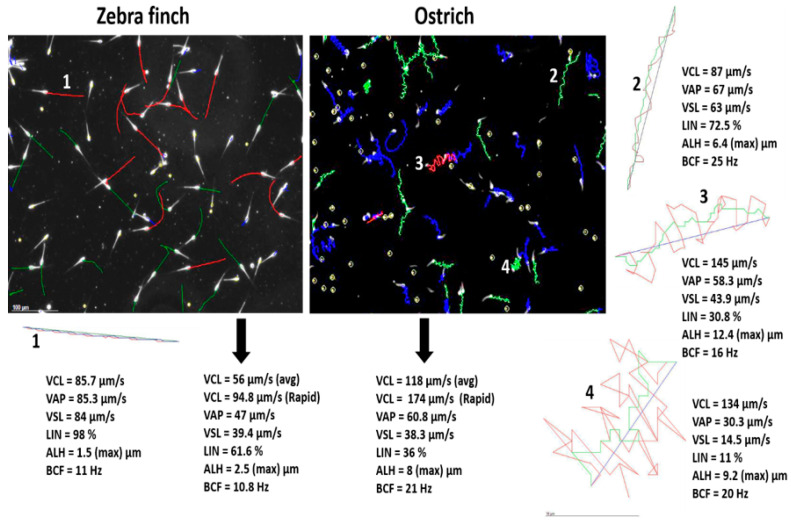
CASA analysis of sperm of two bird species, Zebra finch (Passerine, Neognathae) and ostrich (Paleognathae). In Zebra finch, great similarity can be observed in the form of sperm tracks, contrasting the large variation found in ostriches which show less forward-progressing sperm. Sperm 1 is representative of a rapid progressive sperm from Zebra finch and 2, 3 and 4 three are different swimming patterns of ostrich sperm, with 4 showing typical features of a hyperactivated sperm.

**Figure 4 animals-11-01491-f004:**
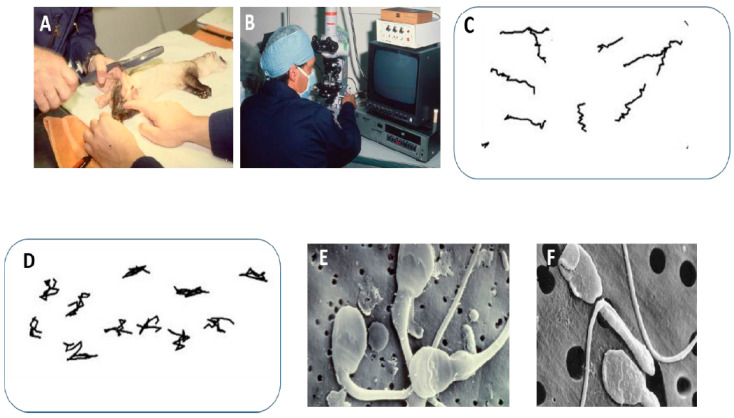
(**A**) Examination of black-footed ferret (BFF) during electrostimulation. (**B**) Part of recording of sperm motility of the nine founder males. Detailed video recordings performed of sperm in cryopreservative medium before cryopreservation. (**C**) Sperm tracks of BFF sperm in TEST medium and (**D**) in equine extender causing hyperactivation and typical hyperactivated tracks. (**E**) Scanning electron micrograph of three normal Siberian ferret sperm (two still immature and retaining a cytoplasmic droplet). (**F**) BFF sperm showing multiple abnormalities, such as lipped acrosome and midpiece with Dag-defect. In BFF (Black footed ferret) and Siberian ferret, the posterior part of the acrosome is scallop-shaped.

**Figure 5 animals-11-01491-f005:**
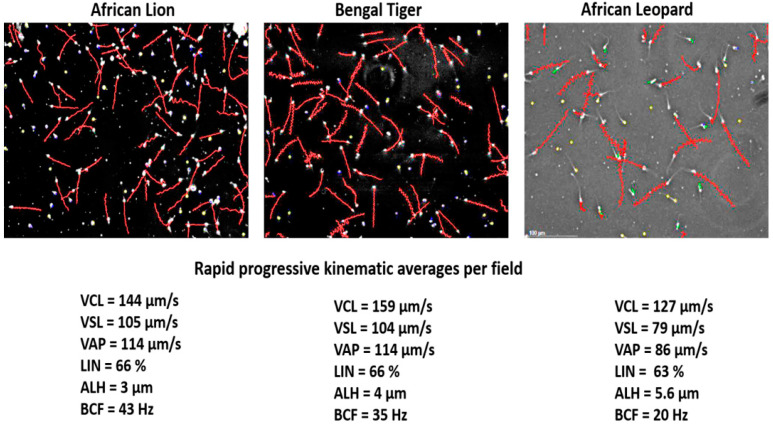
CASA analysis of selected big cats, showing SCA captured fields for the African lion, Bengal Tiger and African leopard. Despite species specific traits, there is a remarkable similarity in the sperm motility patterns (rapid progressive sperm) and accordingly, narrow ranges for the kinematic values among the three species. Not shown: wild dog and cheetah fall within the same ranges for kinematic values.

**Figure 6 animals-11-01491-f006:**
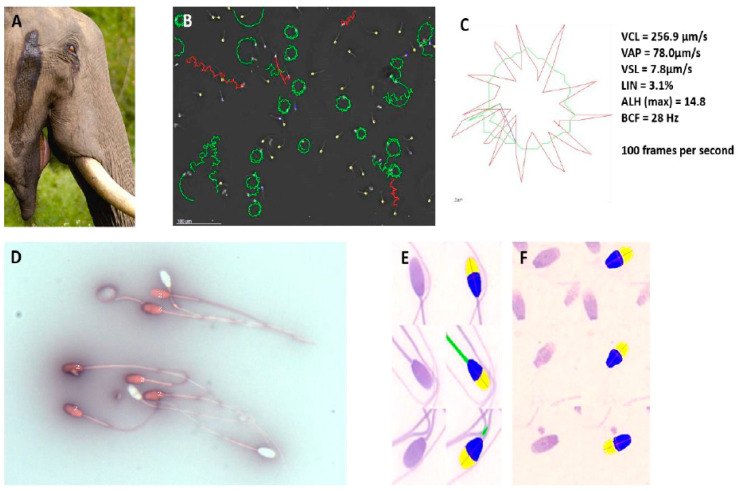
(**A**) Indian elephant in musth. Notice typical oil secretion along the side of the head. (**B**) CASA-captured field with high percentage of hyperactive African elephant sperm. (**C**) Typical star spin hyperactive sperm track of African elephant showing six kinematic parameters. (**D**) Nigrosin-eosin vitality smear of African elephant sperm showing dead sperm in pink and live sperm in white. (**E**) CASA morphology analysis of African elephant sperm using the SCA (Sperm Class Analyzer) system of Microptic S.L. (Barcelona, Spain). A large number of morphometric parameters, as well as the percentage abnormality, can be derived from such an analysis. Yellow, acrosome; blue, rest of head; green, midpiece. (**F**) African elephant sperm after long-term contraceptive treatment with Improvac. Sperm heads with detached tails are a typical feature of this treatment [46].

**Figure 7 animals-11-01491-f007:**
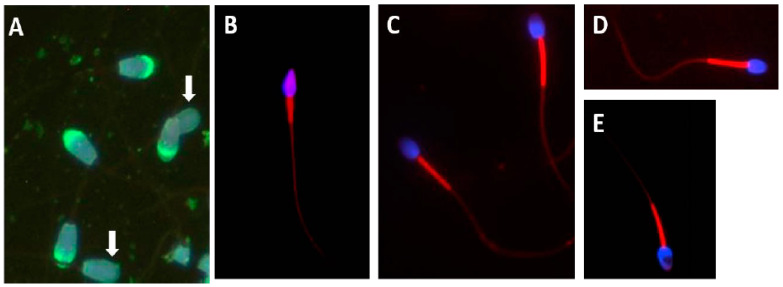
(**A**) Tankwa goat sperm showing intact acrosomes (green fluorescence—FITC-PNA) and two sperm without acrosomes (white arrows). (**B**–**E**) Examples of mitochondrial membrane potential assessment, with midpieces fluorescing red and Hoechst staining in blue for nucleus. (**B**) Human sperm, (**C**) Chacma baboon, (**D**) Vervet monkey, (**E**) Rhesus monkey, from Maree [64].

**Figure 8 animals-11-01491-f008:**
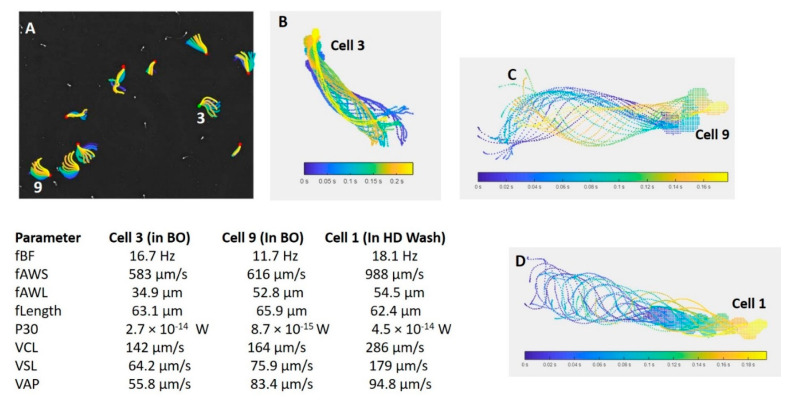
Flagellar and Sperm Tracking (FAST) showing flagellar analysis of vervet monkey sperm at 169 fps. (**A**) Basic flagellar patterns of vervet monkey sperm in a BO medium containing caffeine to induce hyperactivation. (**B**,**C**) Sperm cell 3 and 9 from (**A**), as indicated in white numbers. (**D**) Sperm cell 1 in HD-wash medium, which contains substances such as myo-inositol that can produce capacitation changes. (**B**–**D**) Five flagellar analysis parameters and three CASA, head centroid parameters, as indicated on the left.

**Table 1 animals-11-01491-t001:** The most important macroscopic, microscopic and biochemical parameters for defining sperm quality; CASA = Computer Aided Sperm Analysis.

Parameter	Methodology/Comments
Macroscopic and physical	
Volume in mL	Most accurate by weighing the sample
Liquefaction time	
Colour (white, cream, brown, grey)	Subjective visual inspection
Viscosity	Time to fill a 20 μm deep Leja chamber
Mass movement (scale e.g., 1 to 5)	Subjective visual inspection
pH	pH meter, Panpeha pH indicator strips
Microscopic	
Percentage motility	Manual or CASA
Percentage progressive motility	Manual or CASA
Percentage normal morphology	Manual or CASA
Percentage vitality (live/dead)	Manual or CASA
DNA Fragmentation	Manual or CASA
Immunobead test	Manual
Mixed antiglobulin reaction test	Manual
Biochemical	
Zn, Fructose, alpha glucosidase	

**Table 2 animals-11-01491-t002:** Microscopic, biochemical and molecular biology parameters relating to sperm functionality. VCL = curvilinear velocity; VSL = Straight line velocity, VAP = Average path velocity; CASA = Computer aided sperm analysis; TEM = Transmission electron microscopy.

Parameter	Methodology/Comments
Microscopic	
Sperm concentration/mL and number in ejaculate	CASA, spectrophotometric, nucleocounters
Motility (percentages and sub-populations)	CASA
Percentages rapid and medium progressive	CASA
Slow non-progressive	CASA
Rapid, medium and slow swimming speed	CASA
Number of sperm for each of above % per	CASA
ejaculate volume rather than per ml	
CASA kinematic values (VCL, VSL, VAP for	
sub-populations such as Rapid, medium, slow)	
Sperm mucous penetration per ejaculate	CASA
Hyperactivation	CASA
Flagellar analysis	FAST
Percentage vitality	CASA
Percentage hypo-osmotic swelling	Manual
Fragmentation	CASA
Percentage normal sperm morphology with	CASA
emphasis on indices (TZI and MAI) and sperm	CASA
morphometrics (length, width, perimeter, etc.)	CASA
Acrosome intactness and acrosome reaction	CASA
Reactive oxygen species (ROS) (Fluorescence)	
Mitochondrial membrane potential (MMP) Biochemical	
Energy measurements: ATP, AEC and O_2_ consumption	
Molecular biology	
Centriolar development	Fluorescence, TEM
Telomere length	PCR
Proteomics	
Metabolomics	

**Table 3 animals-11-01491-t003:** Data summarized from Table 3 by Gallego and Asturiano [13] Ranges for minimum and maximum averages have been indicated; VCL = Curvilinear velocity.

Teleost Fish Groups	Percentage Motile Sperm	VCL (µm/s)
	Minimum and maximum averages	
Freshwater		
Representing 3 families	80–95	135–336
and 19 species		
Marine		
Five species	75–95	110–300

**Table 4 animals-11-01491-t004:** Self-explanatory table of a comparison of selected sperm traits of the five vertebrate classes.

Sperm Trait	Pisces (Teleosts)	Amphibia (Anura)	Reptilia	Aves	Mammalia (Selected)
Sperm concentration	Very high > 10 billion/ml	170 M–1.5 billion/mmillion/mL	Too little information	High, variable (commonly > 1billion/mL	Low to high (40 M–3 billion/mL)
Sperm structure	Small simple, round	Filiform head, small	Filiform, longest	Filiform head, large mid-	Very short to long
	to oval head, small	midpiece, few	sperm among verte-	piece, few mitochondria	heads in rodents,
	Midpiece (3–5 mitochondria, flagellum	mitochondria	brates, longest midpiece	long midpiece	midpiece small with 6
	simple axoneme (9 + 2)	Often undulating	piece, 9 + 9 + 2 tail	9 + 9 + 2 tail	to 300 mitochondria
		Membrane 9 + 2 axoneme			Tail 9 + 9 + 2 tail
	No acrosome	Acrosome cap	Acrosome cap	Small acrosome	Acrosome cap 40 to 70% head cover
Normal Sperm morphology (%)	High, > 80	High, > 80	High, > 80	Variable (low to high)	Variable (low to high) 15–90
Sperm Motility (%)	High, > 75%	13–60	Too little information		Low to high (15 to 90)
Sperm Progresson (%) (STR)	High, > 80	High, >80	High, 80	Variable (75–95)	Variable (15–90)
Sperm Velocity (VCL, µm/s)	110–350	15–60	60–80	70 to 250	(60–400)
Sperm Linearity	High in marine, >80	40–75	60–80	30–98	Low to high (40–85)
Vitality (%)	>80	>80	>80	>80	15–95
DNA fragmentation (%)	Mostly < 10	Too little information	Too little information	Mostly < 5	Mostly < 10

## Data Availability

Not applicable. The author has read and agreed to the published version of the manuscript.

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
