# Peer review of "Status of Sperm Functionality Assessment in Wildlife Species: From Fish to Primates"

_animals, 2021, doi:10.3390/ani11061491_

Round 1
Reviewer 1 Report
I thank the author for addressing all my (minor) points, and particularly for the inclusion of summary tables. I am therefore pleased to recommend this work for publication.
Author Response
I appreciate the reviewer's comments and suggesting that the paper is now in order as it stands for publication.
Reviewer 2 Report
I have again spend hours reading your manuscript. Some positive changes have been made. It still lacks a clear message. It is very long and many of the supporting paragraphs do not clearly support the introductory statement of purpose. Many examples are vague or inconclusive. You felt that my comments last time were "destructive." They were honest comments without an effort to be soft. After spending hours reviewing a manuscript that had so many basic errors in structure and design, I did not have patience for tiptoeing around. I felt you needed to hear the honest opinion of a reviewer who has read hundreds and hundreds of research and review articles, and published dozens himself.

Author Response
I appreciate the in depth analysis of the reviewer of my paper.
Commented [A1]: Should be evident that aquaculture is fish
Author Response (AR): There are other animals which aquaculture refer to even salamanders (Amphibia)
Commented [A2]: Seems myopic to focus on black-footed ferret when 100’s of endangered
species are involved in captive breeding programs, especially in an article covering such a wide
AR: The question is either a list of major endangered species and then discussion of some. Here I selected the BFF as it was by 1988 when this work was performed the most endangered mammalian species ever and also it is one where some CASA applications were done. What would you gain by adding a big list or discuss many examples? It is the principle that is relevant and the application of the technology to wildlife.
Commented [A3]: Connfusing. One important
approach for what? For understanding
reproductive physiology? taxonomic base.
AR: I have stated clearly to better understand sperm physiology on a comparative basis
Commented [A4]: This is your stated purpose of
this manuscript and so everything in it should
support this statement.
AR: But I have demonstrated the use of these technologies throughout
Commented [A5]: How are these two separate
methodologies. Isn’t flagellar analysis a
component of CASA?
AR: Both use computer software but they are vastly different. CASA deals with head centroid movement while FAST deals with flagellar kinematics at high frame rates and totally different principles and interpretations are employed.
Commented [A6]: How are these parameters
addressed in your review?
AR: I have shown examples of Vitality and morphology (elephant, Fig. 6 E – F) as good examples.
Deleted: to 46, AR: Accepted
Deleted: deep freeze 47 AR: Accepted
Commented [A7]: This is misleading. It seems
your paper is a call for investigators to use CASA
for semen studies, which certainly is a good
message since so many do still rely on subjective
estimations. But CASA systems have been around
for decades. While they have improved somewhat
over that time, the automated analysis of
concentration and motility is hardly “the latest
technology” and hasn’t been for at least 20 years.
AR: It is not misleading since CASA has not only immensely improved the software dealing with thresholding only sperm (no background particles), only motile sperm and in addition many new modules have been added such as Vitality, Fragmentation, Acrosome reaction and detection of leukocytes. Furthermore, the leading companies in the world improve their system on an ongoing basis and keep on incorporating aspects that make it better and faster and more accurate. Indeed using the latest cutting edge technology!
Commented [A8]: A review paper does not have
methods
AR: was merely alluding to the types of technology employed in the Abstract.
Commented [A9]: These are not addressed in any
appreciable depth in this review.
AR: I have indicated which of these have been addressed but it is also to inform what is available in terms of Methods
Commented [A10]: A review paper does not have
results
AR: This aspect as well as A8 were to put a more structured Abstract together and it should be noticed that these sections do not form part of the body of the review at all. So, indeed the review does not have Methods or Results in the formal sense of the word.
Commented [A11]: Is this really generally true
across the wide number of freshwater and marine species
AR: This comment makes no sense as the blue line connects to Amphibians and birds…
Commented [A12]: You do not clarify that this
means, in contrast to sperm quality; you also use
semen and sperm interchangeably in the
following paragraphs, which they are not
AR: Sperm quality is first treated in the first part from beginning of Introduction. Then sperm functionality is treated from Line 95 (original version). Sperm and semen terms perfectly correctly used. When dealing with motility it is clearly the sperm characteristics and semen refers to both the seminal plasma and sperm.
Deleted: 87 AR: I do not have a line 87
Commented [A13]: Confusing. Is was not the
intention of the WHO to have their parameters
and recommendations reviewed? And actually
your statement is that the users of the manual
have been reviewed themselves…
AR: No I clearly state that WHO meant the manual to be one to evaluate sperm quality/parameters. However, users tend to misuse it for assessment of fertility status. It states nothing more or less.
Deleted: 88
Deleted: 89
AR: Not sure what has been deleted as it is not shown in review pane. I do not have these lines in your original mark-up
Commented [A14]: How do you back up this
statement? Above you just stated that sperm
quality must be distinguished from sperm
fertility.
AR: Added potential sub-fertility
Commented [A15]: How do you back this
statement up and how does it fit in with the rest of
the paragraph?
AR: Relates to the issue of quality versus potential fertility
Commented [A16]: The intent of this statement is
not clear. It is worded as if semen concentration is
something that is shared by some species, as a
parameter of merit. I think you mean that some
species have high concentrations and some have
low. But even this is misleading and not helpful.
In all species there is significant variation between
individuals. And comparisons between unrelated
species is not especially enlightening.
AR: It is clearly stated that while there are large overlaps of many sperm characteristics among several domestic species there is also sperm species specificity. Sperm concentration may vary a lot among individuals of the same species but when trying to define high quality sperm it is based on solid analysis of thousands of bulls with good conception records. Comparison among different species is of great value and has many applications such as in sperm competition studies.
Commented [A17]: Citation? This is a true
statement, but you do not provide a citation. And
your next statement is an example to the contrary.
AR: I have produced reference [7]. Also the same reference indicated the contrary when comparing several semen parameters as stated. On one hand a specific set that relates to fertility but then overall parameters not necessarily.
Commented [A18]: Confusing and meandering
statement
AR: Yes, the sentence has been restructured to make sense.
Commented [A19]: Redundant.
AR: Why? I disagree
Commented [A20]: I’m not sure how much it adds
to the discussion to drop these names, but the
treatment should be consistent in whether you
include first names, first initials, and including
“Dr” with Barbara Durrant’s name.
AR: Dr Barbara Durrant added
Commented [A21]: This is complicated, confusing,
and overly ambitious.
AR: Possible but I did present at least one example of each example presented.
Commented [A22]: Within 8 seconds of what?
AR: Added: …after dilution with water/medium..
Ejaculation? How would that even be possible? AR: No no not in fish
Commented [A23]: Don’t you think you should
emphasize the functionality of this type of motility
in facilitating external fertilization? Especially as
you attempt to draw comparisons below with
amphibians and then terrestrial vertebrates with
internal fertilization? And thus make a better case
for better CASA protocols?
AR: I get the point but I think it is broadly covered that these technologies assist in both external and internal fertilizers.
Commented [A24]: How is this part of your
summary? You did not address any of these
aspects in your text.
AR: In the summary I clearly state potential applications that can now be adopted. Cannot see the need to explore each of these particularly in view of the fact that the Reviewer feels the paper is too long.
Deleted: s 292
Commented [A25]: You throw molecular biology
into the conversation here, but do not explore it or
offer any justification. Either leave it out or justify
it.
AR: Molecular biology facet is not a sudden new topic but I am trying to make the point that even more is gained when combining e.g. CASA technologies with other approaches
Commented [A26]: This is an example of what is
common in this paper. I understand that English
is not the author’s first language. The person
employed to edit this manuscript for English
should help simplify the style for clarity. This
sentence meanders around and is very difficult to
understand, leaving the reader lost and confused.
AR: I removed the sentence.
Commented [A27]: None of these points were
made or explored in your section onn amphibians;
why are they your summary? You should be
using these points to support your manuscript
stated focus from your introduction.
AR: I added it here as additional information as I did not want to explore it in the body of the text for Amphibians and again an attempt to rather include information but keep length of the paper in mind. I cannot see why in a summary one only needs to stick to the content of that section and not add useful and important information. After all I have a morphology section for each vertebrate class and this was the best place to add for Amphibians.
Commented [A28]: Out of context and without
any explanation, this statement is not helpful nor
illustrative.
AR; I disagree. As temperature rises there is normally a concomitant increase in cellular activity and vice versa and here it is not the case. So it is unexpected and answers A29
Commented [A29]: Why unexpected? Clarify. AR: See A28
Commented [A30]: What does this mean? Out of
context and not explained.
AR: I was contrasting the issue of sub-population approach to actual values occurring in reptiles and that it differs from crocodiles.
Deleted: Furhtermore 383
Commented [A31]: And those “important”
approaches were what? And discovered what?
Suggesting we should do what in other
investigations? You are dropping vague
references without any meat.
AR: But it clearly refers to the preceding sentence dealing with telomere length and that it also relates to sperm kinematics. Nothing vague about that.
Commented [A32]: But you do not explore these
topics…
AR: But they have been explained in the paragraph just prior to this one.
Commented [A33]: This seems to be your major
take-home point: that scientists should use CASA
to focus on specific parameters of sperm motility.
But you never really state that in your
introduction and you don’t spend enough time
justifying it in your text. The 3-D sperm motility of
teleost fish was a good start, but even there you
stopped short. Also the foam frog was a good
example, but did not investigate how CASA could
help.
AR: I made the CASA approach clear from the very beginning.
Commented [A34]: Not that hard to inseminate.
Did it produce calves?
AR: Yes calves produced. Added this in the same sentence
Commented [A35]: How does this paragraph fit in
with the rest of the manuscript?
AR: CASA was performed to confirm that sterility was reached
Commented [A36]: You need to justify this
statement. What evidence is there that sharing
sperm characteristics (and you don’t name what
those are) means you are closely related as
opposed to orthopedic or skull or dentition or
embryology?
AR: I stated morphology (morphometrics) was an important characteristic and this is so well known known in the literature that sperm morphology relates to phylogeny, taxonomic affinities etc.
Commented [A37]: And do more closely related
species show more closely similar sperm
characteristics? See the comment above.
AR: Yes, and is abundantly described in literature
Commented [A38]: Which were…?
AR: No offspring produced
Commented [A39]: OK…and?
AR: Will add the relationship between good sperm parameters and good health status
Commented [A40]: What does this mean?
AR: Added in CONSECUTIVE ejaculates
Commented [A41]: Importance of this observation
to this paper?
AR: This showed that CASA analysis revealed high quality gametes as evaluated by fragmentation and acrosome reaction analyses.
Commented [A42]: OK. Explain then…
AR: But it is explained directly after your comment
Commented [A43]: Did you discuss this at all?
AR: I have now included a summary section dealing with related aspects but again as it stands it is an important point to make even though it has not formed part of the body of the text. One can surely add information to more comprehensively summarize/understand.
Reviewer 3 Report
The MS is ready to be published. Only minor typographical mistakes have been found.
Line 161. There is at least one fish species where the flagellum has an structure of 9+0 instead the 9+2 structure (Gibbons et al. 1985).
Gibbons, BH, Baccetti, B, Gibbons, IR (1985). Live and reactivated motility in the 9 + 0flagellum of Anguilla sperm. Cell Motil. 5, 333–350
Line 172. Change “Galego” by Gallego.
Line 357. Correct “Furhtermore” by Furthermore
Line 465. Eliminate “re” or put “regarding”
Line 485. Correct Bback-footed ferret
In the table 4, it is indicated in the structure of the flagellum a tail 9+9+2, in reptilia, aves and mammalia; is it a mistake? It is not commented in the text, where reference to the 9+2 flagellum is made at least in aves.
In that table, I would put values, instead of High/Low, in the parameters of Sperm Progression, linearity, etc, in order to compare between different species
Author Response
Reviewer 3 comments
I want to thank the reviewer for all the valid comments below and note my responses below
The MS is ready to be published. Only minor typographical mistakes have been found.
Line 161. There is at least one fish species where the flagellum has an structure of 9+0 instead the 9+2 structure (Gibbons et al. 1985).
Gibbons, BH, Baccetti, B, Gibbons, IR (1985). Live and reactivated motility in the 9 + 0flagellum of Anguilla sperm. Cell Motil. 5, 333–350
Author response: I have modified text stating ….”mostly 9 + 2 pattern” and accordingly does not need to add the reference for such a minor aspect.
Line 172. Change “Galego” by Gallego.
Line 357. Correct “Furhtermore” by Furthermore
Line 465. Eliminate “re” or put “regarding”
Line 485. Correct Bback-footed ferret
Author response: I have corrected all of the above
In the table 4, it is indicated in the structure of the flagellum a tail 9+9+2, in reptilia, aves and mammalia; is it a mistake? It is not commented in the text, where reference to the 9+2 flagellum is made at least in aves.
Author response: It is indeed correct that all internal fertilizers have a 9 + 9 + 2 flagellar pattern but often the end piece of the tail is only 9 + 2 also in Aves
In Table 4, I would put values, instead of High/Low, in the parameters of Sperm Progression, linearity, etc, in order to compare between different species
Author response: Thank you for these useful comments and I now added values that could be considered more representative
Reviewer 4 Report
Congratulations for your work, here are a series of comments with the aim of improving the initial manuscript:
- Figures: Text should be justified.
- Tables: In some titles, sometimes you use bold, others italics. Please unify.
- General comment: Tables and Figures: All tables and figures should be self explanatory, for example VCL (curvilinear velocity) is not defined in Tab 1. Please, make sure that all abbreviations are correctly explained.
- SCA is not defined.
- A script with the abbreviations used would facilitate the understanding of the text.
- CASA is not defined in the text.
- TEM is not defined
- FAST is not defined in the text.
- I think is important the part of “endangered” especies, nevertheless it does not refer to works where this protection or conservation is related to sperm quality.
- Table in two pages.
- Table 2 has an extra space.
- Table 3 in bold.
- Table 3 is in two pages.
- Figure 4b.
- Spaces.
- In other page.
- Add space
- You should not make that type of binliographic references, it would be better to do them on the published works. You use it in several moments.
- Please do not use abbreviations in conclusions.
- I think is better use impersonal format.
- Conclusions: These are not conclusions, I think this section should be rewritten as it does not present a correct conclusion of the work. You put the objective (really this is on line 146), in this section you should answer this question.
- In general, I think there is more relationship between the objective of the work and the review, Table 4, for example, is the most important, and it is not a conclusion, that comparison should be discussed to enrich much more the importance of this revision. You should add a new approach.
Author Response
- Figures: Text should be justified.
- Tables: In some titles, sometimes you use bold, others italics. Please unify.
- General comment: Tables and Figures: All tables and figures should be self explanatory, for example VCL (curvilinear velocity) is not defined in Tab 1. Please, make sure that all abbreviations are correctly explained.
- SCA is not defined.
- A script with the abbreviations used would facilitate the understanding of the text.
- CASA is not defined in the text.
- TEM is not defined
- FAST is not defined in the text.
Author response: Thank you for these useful points to standardize. I have corrected all these technical points above
- I think is important the part of “endangered” especies, nevertheless it does not refer to works where this protection or conservation is related to sperm quality.
Author response: I have indicated the CASA tracks clearly and indicated the CASA related aspects for the ferret species
- Table in two pages.
- Table 2 has an extra space.
- Table 3 in bold.
- Table 3 is in two pages.
- Figure 4b.
- In other page.
- Add space
Author response: I have corrected all the above. It is often difficult to see this in the marked up copy but it is clearly correct in the final corrected copy
- You should not make that type of binliographic references, it would be better to do them on the published works. You use it in several moments.
Author response: Not quite clear what is meant by reviewer. I need to indicate that in many examples used, our group has performed for first time and truly wild/free ranging populations or that we have investigated facets that were not previously studied. However, in most instances this has been backed up by our peer reviewed publications.
- Please do not use abbreviations in conclusions.
Author response: Have remove abbreviations in conclusion
- I think is better use impersonal format.
- Conclusions: These are not conclusions, I think this section should be rewritten as it does not present a correct conclusion of the work. You put the objective (really this is on line 146), in this section you should answer this question.
Author response: I have totally rewritten the Conclusion
- In general, I think there is more relationship between the objective of the work and the review, Table 4, for example, is the most important, and it is not a conclusion, that comparison should be discussed to enrich much more the importance of this revision. You should add a new approach.
Author response: Thank you for this valuable comment: I have now worked into the document a new Section under 8 dealing with summarizing the results on the basis of Table 4.
Reviewer 5 Report
This article presented the sperm functionality assessment from fish to primates in focusing on sperm motility using CASA.
This review is very interesting and valuable data to contribute to understand sperm assays for wildlife species.
This paper is well-written.
Author Response
This article presented the sperm functionality assessment from fish to primates in focusing on sperm motility using CASA.
This review is very interesting and valuable data to contribute to understand sperm assays for wildlife species.
This paper is well-written.
Author response: The reviewer is thanked for the positive comments.
Round 2
Reviewer 2 Report
This paper has much potential and could be a very interesting paper. Of the many in depth comments that were suggested in my previous review, only minor changes have been made. The majority of the suggestions or comments were met with explanations or refutations, without changes being made in the manuscript. I had no previous bias against this paper or author, nor do I have any now, but it seems we have very different opinions about the quality of the paper in its current form.
Reviewer 4 Report
Congratulations, great job.
This manuscript is a resubmission of an earlier submission. The following is a list of the peer review reports and author responses from that submission.
Round 1
Reviewer 1 Report
This review provides a needed overview of sperm functionality across wildlife species. Through the use of select examples, the author highlights important inter-species variation with reference to reliable methods for assessing the relevant quantities. The work is scientifically accurate and accessible to a wide range of expertise. I believe that this manuscript will be a valuable addition to the field and recommend that it is accepted for publication.
I enclose the following (minor) comments:
Page 5: The author highlights the comprehensive review on fish sperm by Cosson et al. 2008. It is worth pointing out the (very) recent further paper by this author on the subject of fish spermatozoa, Cosson 2020, Fish Physiology and Biochemistry as this provides an in-depth and updated view of the subject and its history.
Page 5, L171: The author suggests that static sperm may be excluded with a threshold of VCL < 25 um/s. This seems rather fast to me and I suspect that they may have intended to write 5um/s.
Given the large amount of informative features discussed throughout this work, it may be helpful to the reader to conclude with a brief explanation (or maybe just a table?) summing up some of the key differences in sperm features across the species discussed.
Author Response
This review provides a needed overview of sperm functionality across wildlife species. Through the use of select examples, the author highlights important inter-species variation with reference to reliable methods for assessing the relevant quantities. The work is scientifically accurate and accessible to a wide range of expertise. I believe that this manuscript will be a valuable addition to the field and recommend that it is accepted for publication.
.I enclose the following (minor) comments:
Page 5: The author highlights the comprehensive review on fish sperm by Cosson et al. 2008. It is worth pointing out the (very) recent further paper by this author on the subject of fish spermatozoa, Cosson 2020, Fish Physiology and Biochemistry as this provides an in-depth and updated view of the subject and its history.
Author Response (AR): I appreciate the constructive comments by the reviewer. I have now incorporated the most relevant concepts in latest review by Cosson 2020, Fish and Biochemistry
Page 5, L171: The author suggests that static sperm may be excluded with a threshold of VCL < 25 um/s. This seems rather fast to me and I suspect that they may have intended to write 5um/s.
AR: Initial manuscript page 5. I realize this 25um/s as cut-of for motile sounds high but we have vigorously tested it as best approximation for most species…even human. Also my colleagues in Sheffield found that any bird sperm swimming less than 40um sec just not worthwhile to analyse. One of the criteria we used for immotile…if head seems to make small movements but tail does not vibrate..immotile.
Given the large amount of informative features discussed throughout this work, it may be helpful to the reader to conclude with a brief explanation (or maybe just a table?) summing up some of the key differences in sperm features across the species discussed.
AR: The idea of the Table or Tables assisting to summarize comparing information is a very good idea and I have incorporated a summary document for Pices. For the rest of the Vertebrate groups I have rounded each one off with an improved summary and a final Table. Thank you for this point.
Reviewer 2 Report
The goal of this paper is unclear. This is not a research paper in any sense. As a review paper, the topic to be reviewed is unclear. The title does not reflect the focus of what is presented. The author should decide on a precise goal of what is to be reviewed and them dial in on that purpose.
This paper lacks appropriate discussion of fluorescent markers used to evaluate sperm functionality. There is little to no discussion of DNA integrity, mitochondrial integrity, oxidative stress, membrane permeability, etc
The meaning of many statements are lost in bold or fancy language structure. In the introductory portion, the author uses phrases like "it is evident, "well described," "great success," and things are "of great importance" but they do not simply demonstrate the obvious evidence or the great importance with any simple, concrete examples. It would be more effective to eliminate this hyperbole and simply describe and let the readers decide what is evident, what is successful, and what is of great importance based on the evidence presented by the author. Shorter sentences would greatly benefit this author's style. Much of the paper consists of long, meandering sentences that leave the reader lost and confused by the end. I found myself rereading many of the sentences trying to stay with the author and understand what his point was. Flowery phrases were strung together with the end result being nonsense. Keep it simple. State your point simply and directly. Don't tell the reader how to interpret the data; let the reader judge the importance based on your evidence.
After some confusing statements about what has been done by others, the author states what "the correct approach" (line 104) is...with what evidence? Three papers are cited later in the paragraph (with the author's contributions, so some bias here), but not discussed in the body of the paper to support this bold statement.
The paper lacks any logical progression. The author jumps from one random point to another, often just throwing out pet peeves (use of "sperm density" or sperm morphology analysis) leaving the reader feeling like their head is spinning as they try to follow the narrative. A review paper, like this one, should read in a logical progression and tell the scientific story in a way that the informed reader can easily follow.
The author calls this a "review-cum research paper." How is it in any way, shape, or form a research paper? Where are the methods? What was tested? Where statistics involved? What were the study groups? What was the hypothesis? Where are the results of any tests run? This paper feels like a list of opinions, not a logical, comprehensive, unbiased review and certainly not a research paper. The omission of materials and methods prevents readers (and reviewers) from properly scrutinizing and judging the merit of the data. This flies in the face of all peer-reviewed procedures for scientific publication.
Tables 1 & 2: I can see no logical reason to have these two tables as separate tables. Even as stand-alone tables I do not gain useful information from them.
Line 124 contradicts earlier comments about the importance or lack of importance of certain, isolated sperm parameters by broadly overstating the importance of sperm motility. Motility by itself has been shown to be limited in value in some species when predicting fertility. This seems to be the point that should be a take-home in this paper: each species must have its own analysis before conclusions are made. We can use other species as a foundation and starting place for that investigation, but conclusions must be based on results for that species.
The rest of the paper gives a very broad and shallow or spotty highlight of a few representatives of vertebrate taxa sperm studies. The end point that a large degree of variety in sperm parameters exists is made, but would have been evident to anyone in the field already. The point that objective measurements by CASA systems are better than subjective measures is also something that everyone knows. If the point is to broadly demonstrate variety in sperm morphology and function, a better, more focused, more readable review should be written. If the point is to link sperm parameters with fertility studies, that work has yet to be done in many domestic species, let alone wildlife species.
Author Response
The goal of this paper is unclear. This is not a research paper in any sense. As a review paper, the topic to be reviewed is unclear. The title does not reflect the focus of what is presented. The author should decide on a precise goal of what is to be reviewed and them dial in on that purpose.
Author response: The title has been changed with a better focus on the goal. This is not meant as a research paper on its own but rather predominantly a review with some new research ideas and I have spelled that out; I merely added minimal latest information that is currently in progress. However, most of the concepts have been published. The core of the paper is clearly a review. The goal is clear in the Abstract but I have now tried to also improve the Introduction section to better spell out the goals. I think the reviewer is correct that there needs to be much more direction to the purpose. I believe the information was there but is now hopefully clearly defined.
This paper lacks appropriate discussion of fluorescent markers used to evaluate sperm functionality. There is little to no discussion of DNA integrity, mitochondrial integrity, oxidative stress, membrane permeability, etc
Author response: I made it very clear that the focus is on certain facets of CASA and clearly showed the quantitative approaches in this context and clearly not meant as a comprehensive review on, e.g. fluorescence. However, summary information appears in Table 2 and also facets like acrosome and MMP fluorescence is emphasized in Figure 8.
The meaning of many statements are lost in bold or fancy language structure. In the introductory portion, the author uses phrases like "it is evident, "well described," "great success," and things are "of great importance" but they do not simply demonstrate the obvious evidence or the great importance with any simple, concrete examples. It would be more effective to eliminate this hyperbole and simply describe and let the readers decide what is evident, what is successful, and what is of great importance based on the evidence presented by the author. Shorter sentences would greatly benefit this author's style. Much of the paper consists of long, meandering sentences that leave the reader lost and confused by the end. I found myself rereading many of the sentences trying to stay with the author and understand what his point was. Flowery phrases were strung together with the end result being nonsense. Keep it simple. State your point simply and directly. Don't tell the reader how to interpret the data; let the reader judge the importance based on your evidence.
Author response: English is my second language and I am the first one to admit many shortcomings and there is surely no excuse for this. I have obtained the services of a colleague who is a specialist in editing and who assisted greatly to improve the scientific English, style and flow, and sentence construction throughout.
After some confusing statements about what has been done by others, the author states what "the correct approach" (line 104) is...with what evidence? Three papers are cited later in the paragraph (with the author's contributions, so some bias here), but not discussed in the body of the paper to support this bold statement.
Author response: With all respect all informed CASA users know “the correct approach” re the importance of sperm sub-populations. I referred to it and since I have emphasized this over the last few years in my publications and also cited others, and many citing me, what is the bias? Does the reviewer imply my peer reviewed papers are not authentic? Several of these recent papers have 50 to nearly 200 citations per paper.
The paper lacks any logical progression. The author jumps from one random point to another, often just throwing out pet peeves (use of "sperm density" or sperm morphology analysis) leaving the reader feeling like their head is spinning as they try to follow the narrative. A review paper, like this one, should read in a logical progression and tell the scientific story in a way that the informed reader can easily follow.
Author response: I believe the logical progression has now been considerably improved. The order of some paragraphs have been changed to make the flow better. I have also tried to summarize the sperm functionalities at the end of each of the taxa better.
The author calls this a "review-cum research paper." How is it in any way, shape, or form a research paper? Where are the methods? What was tested? Where statistics involved? What were the study groups? What was the hypothesis? Where are the results of any tests run? This paper feels like a list of opinions, not a logical, comprehensive, unbiased review and certainly not a research paper. The omission of materials and methods prevents readers (and reviewers) from properly scrutinizing and judging the merit of the data. This flies in the face of all peer-reviewed procedures for scientific publication.
Authors response: I have above clearly indicated the position that some incidental research data have been incorporated and I have removed the cum-research and my apologies that this might have been misleading. Furthermore, the “incidental” data incorporated really only applies to some Figures and relate to published research.
Tables 1 & 2: I can see no logical reason to have these two tables as separate tables. Even as stand-alone tables I do not gain useful information from them.
Authors response: Table 1 is what is typically used in many textbooks to indicate which parameters together constitute the definition of sperm quality such as WHO 5 (for human sperm) and those presented by Chenoweth and Lorton in their Andrology Textbook (for domestic animal sperm). Table 2 makes the important distinction that it does not just relate to sperm parameters per se but more to sperm functionality. In the Introduction of the review this aspect is explained in great detail and remains an important component of this review.
Line 124 contradicts earlier comments about the importance or lack of importance of certain, isolated sperm parameters by broadly overstating the importance of sperm motility. Motility by itself has been shown to be limited in value in some species when predicting fertility. This seems to be the point that should be a take-home in this paper: each species must have its own analysis before conclusions are made. We can use other species as a foundation and starting place for that investigation, but conclusions must be based on results for that species.
Authors response: I agree with reviewer but whether motility has either positive or negative connotations it remains a well quantifiable characteristic to assist to make such judgements. Furthermore, while there may be exceptions to motility of less importance to predict fertility in a few species, the majority of papers certainly show that motility, particularly when quantified using CASA is of prime importance and still remains a cornerstone to define functionality. I have also made it clear it is one of…not the only one! So, there is no contradiction.
The rest of the paper gives a very broad and shallow or spotty highlight of a few representatives of vertebrate taxa sperm studies. The end point that a large degree of variety in sperm parameters exists is made, but would have been evident to anyone in the field already. The point that objective measurements by CASA systems are better than subjective measures is also something that everyone knows. If the point is to broadly demonstrate variety in sperm morphology and function, a better, more focused, more readable review should be written. If the point is to link sperm parameters with fertility studies, that work has yet to be done in many domestic species, let alone wildlife species.
Authors response: I disagree that the paper is shallow and spotty. It certainly highlights groups of taxa like what is commonly found in orders of amphibians or mammals by means of some examples. I have in each class covered a broad range of examples and see for example the Table 3 under Pisces. The order Anura has been covered with representative examples. Under Aves I summarized a paper dealing with 47 species and important papers dealing with sperm competition. I made very clear statements with very good examples of the two Super Orders of Aves. Among Mammals I highlighted six diverse orders with many relevant examples such as endangered species.
The points that a variety of sperm parameters is evident (known) to anyone in any way as well as the comment about CASA is surely totally out of context. Basically 80 to 90% of information in a review is known and the idea is to put this together in a broader and understandable context or a new hypothesis. Yes, that these aspects can be considerably improved in being more systematic I agree and I have gone to great trouble to improve these aspects.
Concluding remarks re the reviewer comments:
I remarked that several of the pointers provided by Reviewer 2 above have been used to improve the paper. However, in my 40 years as scientist with more than 150 peer reviewed papers, chapters, reviews and being Associate Editor of a High quality Journal, I have but never ever encountered such a destructive review. Yes, many of the criticisms are surely warranted but it is the tone that is destructive.
Reviewer 3 Report
The MS covers a huge amount of studies, but it is cumbersome, as studies are presented separately, one after the other, and not connected between them. Excessive number of particular names of scientist and the historical approach are noted. It would be more useful, in my opinion, made this review by topics rather than by taxonomical groups. Then, starting for instance with the topic of the spermatozoa shape, and compare between species; later, the topic of sperm velocities compared between species, etc. Tables comparing sperm velocities and other parameters will be useful.
Author Response
The MS covers a huge amount of studies, but it is cumbersome, as studies are presented separately, one after the other, and not connected between them. Excessive number of particular names of scientist and the historical approach are noted. It would be more useful, in my opinion, made this review by topics rather than by taxonomical groups. Then, starting for instance with the topic of the spermatozoa shape, and compare between species; later, the topic of sperm velocities compared between species, etc. Tables comparing sperm velocities and other parameters will be useful.
Bottom of Form
© 1996-2021 MDPI (Basel, Switzerland) unless otherwise stated
Author response: I appreciate the constructive comments by Reviewer 3 and believe the comments are very useful to restructure the paper. While I understand the problem of connecting many diverse taxa by a sperm characteristic approach have problems of its own. However, the problem relates to the fact that in many instances sperm characteristics such as sperm structure, sperm motility and sperm competition aspects are related for a specific taxon. It accordingly becomes difficult to relate these without discussing it within the context of either an Order or a vertebrate class. Furthermore, it is quite problematic to list for example seven or eight key sperm functionalities and within each discuss all the major taxa.
Accordingly, in order to overcome the “connectedness” problem I have tried to summarize as much data as possible (e.g.,Table 3 for Pisces) and better summarize each taxon in terms of diverse sperm characteristics at the end of each vertebrate class. I have paid a lot of attention and made many changes to improve the flow of each section. To this effect I have removed, changed, and inserted new paragraphs. In addition, I tried to provide a more comprehensive conclusion which provides better cohesion or connectedness between the vertebrate classes as well as a self-explanatory table 4 .
I have also removed most of the names and rather retained reference numbers. However, I need to point out that the historical approach is extremely important. Many young investigators reading this may not connect for example the endangered BFF saga without the history.
I have obtained the services of a colleague who is a specialist in editing and who assisted greatly to improve the scientific English, style and flow.